# Nanoscale multistate resistive switching in $WO_3$ through scanning probe induced proton evolution

Fan Zhang [1,2], Yang Zhang [1], Linglong Li[1], Xing Mou[3], Huining Peng[1], Shengchun Shen [1], Meng Wang [4], Kunhong Xiao[1], Shuai-Hua Ji [1,5], Di Yi [6], Tianxiang Nan [3,7], Jianshi Tang [3,7] & Pu Yu [1,5] ✉

Multistate resistive switching device emerges as a promising electronic unit for energy-efficient neuromorphic computing. Electric-field induced topotactic phase transition with ionic evolution represents an important pathway for this purpose, which, however, faces significant challenges in device scaling. This work demonstrates a convenient scanning-probe-induced proton evolution within $WO_3$, driving a reversible insulator-to-metal transition (IMT) at nanoscale. Specifically, the Pt-coated scanning probe serves as an efficient hydrogen catalysis probe, leading to a hydrogen spillover across the nano junction between the probe and sample surface. A positively biased voltage drives protons into the sample, while a negative voltage extracts protons out, giving rise to a reversible manipulation on hydrogenation-induced electron doping, accompanied by a dramatic resistive switching. The precise control of the scanning probe offers the opportunity to manipulate the local conductivity at nanoscale, which is further visualized through a printed portrait encoded by local conductivity. Notably, multistate resistive switching is successfully demonstrated via successive set and reset processes. Our work highlights the probe-induced hydrogen evolution as a new direction to engineer memristor at nanoscale.

Multistate resistive switching devices have been recognized as a promising solution to mimic the functionality of synapses for artificial intelligent computing[1,2]. Various mechanisms have been developed, over the past few decades, to manipulate the electronic states of materials in terms of resistivity, using the charge, spin and lattice degrees of freedom in ferroelectric[3–7], ferromagnetic[8–10], and phase change memories[11], respectively. Among those studies, electric-field controlled ionic evolution has emerged as an appealing strategy for multistate resistive switching. Compared with other mechanisms, ion migration inspiringly stimulates the fundamental physics of signal processing in biological neural systems, greatly extending the range and accuracy in tailoring multiple resistive states in functional materials. With extensive research endeavors, high-performance artificial synaptic devices have already been demonstrated using a wide range of transition metal oxides (TMO), e.g., $WO_3$[12–14], $VO_2$[15], $MoO_3$[16], $SrFeO_{2.5}$[17], and $SrCoO_x$[18,19], where the strong correlation between the

[1]State Key Laboratory of Low Dimensional Quantum Physics and Department of Physics, Tsinghua University, 100084 Beijing, China. [2]State Key Laboratory of Information Photonics and Optical Communications & School of Science, Beijing University of Posts and Telecommunications, 100876 Beijing, China. [3]School of Integrated Circuits, Beijing National Research Center for Information Science and Technology (BNRist), Tsinghua University, 100084 Beijing, China. [4]RIKEN Center for Emergent Matter Science (CEMS), Wako 351-0198, Japan. [5]Frontier Science Center for Quantum Information, 100084 Beijing, China. [6]State Key Laboratory of New Ceramics and Fine Processing, School of Materials Science and Engineering, Tsinghua University, 100084 Beijing, China. [7]Beijing Innovation Center for Future Chips (ICFC), Tsinghua University, 100084 Beijing, China. ✉e-mail: yupu@mail.tsinghua.edu.cn

ionic evolution and the charge and lattice degrees of freedom provides a feasible playground to manipulate the resistive states for artificial synapses[20–22].

To obtain electric-field tunable multistate resistive switching through ionic evolution, three-terminal device architecture was widely employed, in which the ionic liquids and solid electrolytes serve as the gate electrode to shuttle the ions (e.g., $O^{2-}$, $H^+$, $Li^+$, $Na^+$) between the electrolyte and functional materials[12–14]. Such devices have the unique advantage of continuously tuning the electronic properties of functional layers through controllable ionic intercalation and extraction. However, they suffer inevitable limitations of large device footprint and complicated electrochemical reactions, which seriously constrain the device scaling. To address this challenge, new ionic manipulation methods have been developed for resistive switching, in which a voltage-biased scanning probe was employed to control the ionic evolutions[23–25]. The concept of scanning-probe memories has drawn much research attention over the last few decades because of their intrinsic high-density storage capabilities and their unique ability to write and read in an efficient manner[26,27]. This approach not only inherits the essential merit of nano-scaled resolution from scanning-probe-patterned memories, but also provides a convenient pathway to achieve multistate conductivities[28,29]. Oxygen vacancies within the complex oxides become the workhorse for this purpose, which is however entangled with structural defects and lower device reliability[23–25]. Compared with oxygen ion, proton has a much smaller ionic radius that benefits its diffusion in many TMOs in terms of speed and energy consumption[30,31]. A recent study employed protons as the shuttling ion and demonstrated a reversible insulator-to-metal transition (IMT) in $VO_2$ thin films through the electric-field-induced hydrogenation[32]. Unfortunately, an elevated temperature was required in that study in order to boost the ionic incorporation, while modulation of multiple resistance states at room temperature is one of the indispensable requirements for practical applications.

Here, we identified $WO_3$ as a promising candidate to demonstrate the memristive functionality at nanoscale through probe-induced hydrogenation. $WO_3$ is a band insulator with $5d^0$ electron configuration, in which the electron doping through oxygen vacancy, proton and lithium-ion evolution can readily introduce a pronounced metallic state[33]. This material holds promising applications for memories, smart windows and sensors based on such switching[31,34,35]. This work demonstrates a scanning-probe-induced reversible IMT within $WO_3$ at ambient temperature through proton evolution. This transition shows prominent spatial resolution and reliability as evidenced by the probe-patterned portrait and reproducible multistate resistance modulation. This study not only lays the foundation to exploit the scanning-probe-induced proton evolution as a promising strategy to build functional devices, but also offers new possibilities to investigate electronic states in a wide selection of hydrogenated materials[36–38].

## Results

High-quality $WO_3$ thin films (with a thickness of 50 nm) were grown on (001)-oriented Nb-doped $SrTiO_3$ (Nb: STO, 0.7% wt. doped) substrates with the pulsed laser deposition (PLD) method (see Experimental Section and Supplementary Fig. 1a). Figure 1a schematically illustrates the working principle of the probe-induced hydrogen intercalation. In this approach, the Pt-coated scanning probe serves as a movable nano-sized catalyst, assisting the splitting of hydrogen molecules (from the 5% $H_2$/95% Ar forming gas) into protons. Then the positively biased probe further facilitates the proton incorporation between the probe and sample through the potential gradient. In this study, we employed

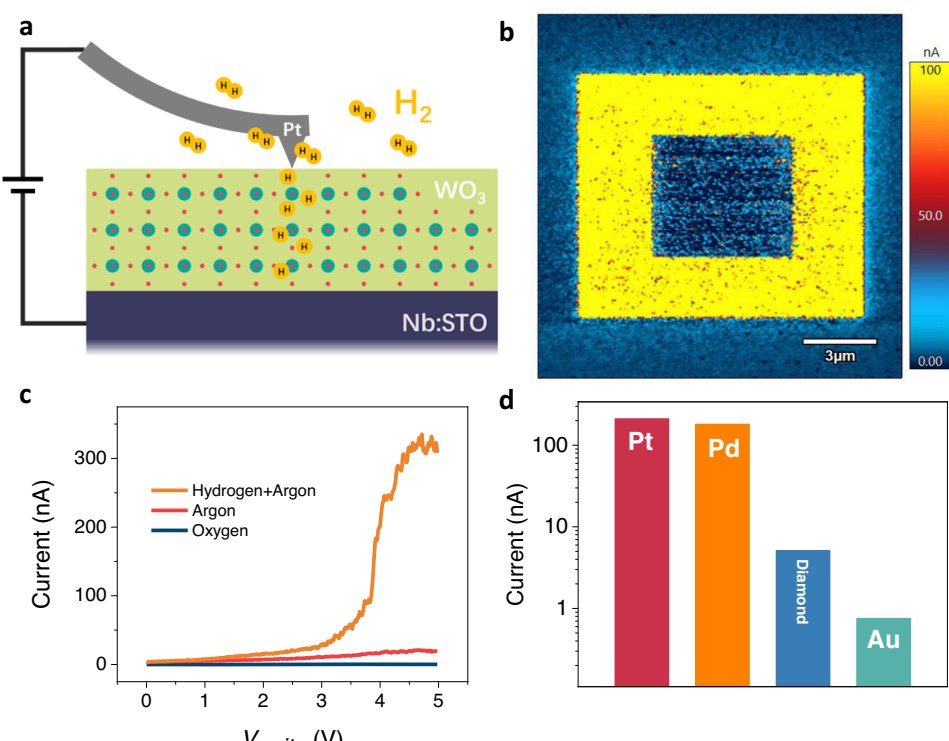

**Fig. 1 | Biased probe induced hydrogenation into $WO_3$ thin films. a** Schematic illustration of the scanning-probe-induced hydrogenation. In this setup, a Pt-coated probe biased with a positive voltage scans the sample surface, and the whole process is carried out within a hydrogen-containing gas environment.
**b** Conducting AFM mapping of a hydrogenated and dehydrogenated pattern, which were written and erased with + 4 and −4 V, respectively. The reading voltage is 0.1 V. The corresponding topography of the scanned region is demonstrated in Supplementary Fig. 1. **c** Voltage-dependent local currents for regions formed at different gas environments with identical Pt-coated probes. **d** Comparison of hydrogenation effectiveness with various conducting tip with different coating layers: Pt, Pd, doped diamond and Au. The data records the maximum conductivity change shown in Supplementary Fig. 5.

the highly doped Nb:STO substrates[39–43] as a bottom electrode to facilitate the tip-induced hydrogenation as well as subsequent conducting atomic force microscope (cAFM) measurements. We first investigated whether the probe-induced proton evolution method is compatible with the WO$_3$ thin films at ambient temperature (i.e., 25 °C for the current study). For this purpose, we carried out a double-switch experiment, in which a 10-μm-by-10-μm square was initially written with a voltage of 4 V, and then a 5-μm-by-5-μm square was written with a voltage of −4 V at the center. To directly visualize the pattern, we measured the scanned region with the cAFM technique as shown in Fig. 1b. Dramatically enhanced electrical conductivity is observed at the positive bias scanned region as compared with the pristine region, while it can be nicely reversed back with the negatively biased scan. Figure 1b shows a uniform contrast of the entire hydrogenated area through the cAFM measurements, which is distinct from conductive filaments-induced memristive switching, where an inhomogeneous current map would be observed[44]. Notably, the sample morphology remains unchanged after the hydrogenation-dehydrogenation cycle (Supplementary Fig. 1b), indicating that the crystallinity is not deteriorated, which is essential to achieve reliable and reversible IMT. The topographic measurements also demonstrate a clear lattice expansion effect at the hydrogenated region, which will be further discussed later.

In the probe-induced proton evolution scenario, the forming gas serves as the hydrogen source, and the probe is employed as a catalyst to facilitate the reaction. As it was demonstrated that the oxygen vacancy accumulation could also lead to an IMT in WO$_3$ through electron doping[45], we subsequently carried out a series of control experiments to discriminate the contribution of oxygen ions and protons. We first patterned the sample with a gradually increased positive biased voltage at different gas environments and then probed the same regions through cAFM as shown in Supplementary Fig. 2. Figure 1c summarizes the results into a writing voltage dependence of the local currents. The region written within forming gas shows the most pronounced conducting level (hundreds of nA) among all these measurements. This value is much higher than that observed in the conducting domain walls in ferroelectrics[28], making it easier to access the state at a small voltage. Importantly, the previous study reveals a critical voltage of 4 to 5 V to induce IMT in the VO$_2$ sample[32], while the current study (Supplementary Fig. 2) demonstrates a pronounced contrast at about 3 V. Such a small critical voltage is essential to improve energy efficiency and avoid device failure. In stark contrast, the sample written in Ar gas only shows measurable current at the level of ~10 nA, which should be attributed to the oxygen vacancy formation. This is further supported by the fact that the sample written in oxygen gas ambient remains insulating even with the largest voltage (up to 5 V) applied, where the oxygen vacancy formation is strongly suppressed. Furthermore, within pure Ar gas, the oxygen vacancy-induced IMT is difficult to reverse (Supplementary Fig. 3) because there is no enough oxygen in the environment to compensate for these oxygen vacancies. Although erasing the conducting state is possible in the oxygen environments, the conducting state shows poor retention properties (Supplementary Fig. 4), since the oxygen vacancies would be readily compensated by the oxygen gas. Meanwhile, the retention test in forming gas shows that the hydrogenated area sustains a high conductivity state up to 300 min after the hydrogen intercalated. To clarify the critical role of scanning probe, we also carried out the writing experiments using conducting probes with different coatings (Supplementary Fig. 5), and the obtained conducting currents are summarized in Fig. 1d. We note that noble metals of Pt and Pd are the most widely employed catalysts for hydrogen spillover[46]. However, compared with Pd, Pt has smaller activation energy for the catalytic reaction, meaning it can facilitate hydrogen spillover in a more efficient manner[47], leading to a more pronounced conducting state with identical writing conditions. In contrast, neither Au-coated nor doped

diamond probes can introduce obvious changes in local conductivity. Therefore, these controlled experiments provide compelling evidence to highlight the essential nature of protons (hydrogen ions) for the obtained IMT.

To provide direct evidence for the hydrogenation in the conducting WO$_3$ samples, we carried out the secondary ion mass spectroscopy (SIMS) analysis on hydrogenated and as-grown areas. Before the measurement, the whole sample was capped with Au through sputtering to reduce the hydrogen release. As shown in Fig. 2a, the switched (conducting) area shows much higher hydrogen intensity within the WO$_3$ layer compared with the as-grown (insulating) area. This result also indicates that the hydrogen concentration is higher near the surface the film, which might be related to the intrinsic protonic diffusion process.

We note that ionic evolution is sometimes associated with considerable structural deformation, which might lead to performance degradation[48]. To characterize the evolution of atomic crystalline and electronic structures, we carried out scanning transmitted electron microscopy (STEM) measurements for both the pristine and hydrogenated samples. As shown in Fig. 2b, the high-quality crystalline structure and sharp hetero-interface can be nicely identified at the hydrogenated regions, demonstrating the excellent compatibility of hydrogenation for WO$_3$. To further reveal the role of hydrogenation, we probed the lattice and electronic structures of WO$_3$ before and after hydrogenation. As summarized in Fig. 2c, the hydrogenated sample shows a notable lattice expansion (about 2.3%) along the c-axis, while the in-plane lattice constant remains unchanged due to the epitaxial strain. This structural expansion is consistent with the previous results obtained with the ionic liquid gating-induced hydrogenation into WO$_3$[33]. As for the electronic structure, the intercalation of hydrogen would induce electron doping in materials, which can be characterized by the variation of oxygen K edge of the EELS signals. As shown in Fig. 2d, three featured peaks (A, B, and C) in the oxygen K edge are attributed to the transition from the oxygen 1s state to the hybridized oxygen 2p and tungsten 5d/6sp states. The apparent suppression of the A peak in the hydrogenated sample indicates the presence of more valence electrons within tungsten 5d orbitals[49], which is consistent with the electron doping into the WO$_3$ system.

Notably, the STEM studies also provide a probe to clarify the contribution of oxygen vacancies through integrated differential phase contrast (IDPC) imaging, whose contrast approximately correlates with the potential[50]. Thus, this technique is very effective in quantifying the density of light atomic column like oxygen[51,52]. Figure 2e shows an IDPC image acquired from the hydrogenated area, where the position and intensity of the oxygen column can be accurately determined (Supplementary Fig. 6). After normalizing the intensity of tungsten, no measurable change is observed for the oxygen column before and after probe scanning (hydrogenation), indicating that negligible oxygen vacancy is involved during the IMT.

With the understanding of the underlying mechanism for the probe-induced IMT, we then further evaluate the functional performance of the resistive switching. To demonstrate the switching speed of this hydrogenation process, we carried out writing experiments with a series of scan rates, through which the local dwelling time is systematically varied. The scan rate is a tunable parameter for AFM measurement, which corresponds to the speed at which the AFM probe scans the sample surface. The hydrogenation reaction time can then be approximately estimated as the time used for the tip scanning through a single point, and it can be calculated by the equation of $\frac{P_0}{f_0} \times \frac{S_{tip}}{S_{scan}}$, where $P_0$ is the number (128) of scan lines along the slow-scan direction, $f_0$ is the scan rate along the fast-scan direction, $S_{tip}$ is the estimated contact area (~400 nm$^2$) between the probe and sample surface and $S_{scan}$ is the area of the pattern. As shown in Fig. 3a, the slower scan rates result in a larger current, indicating more

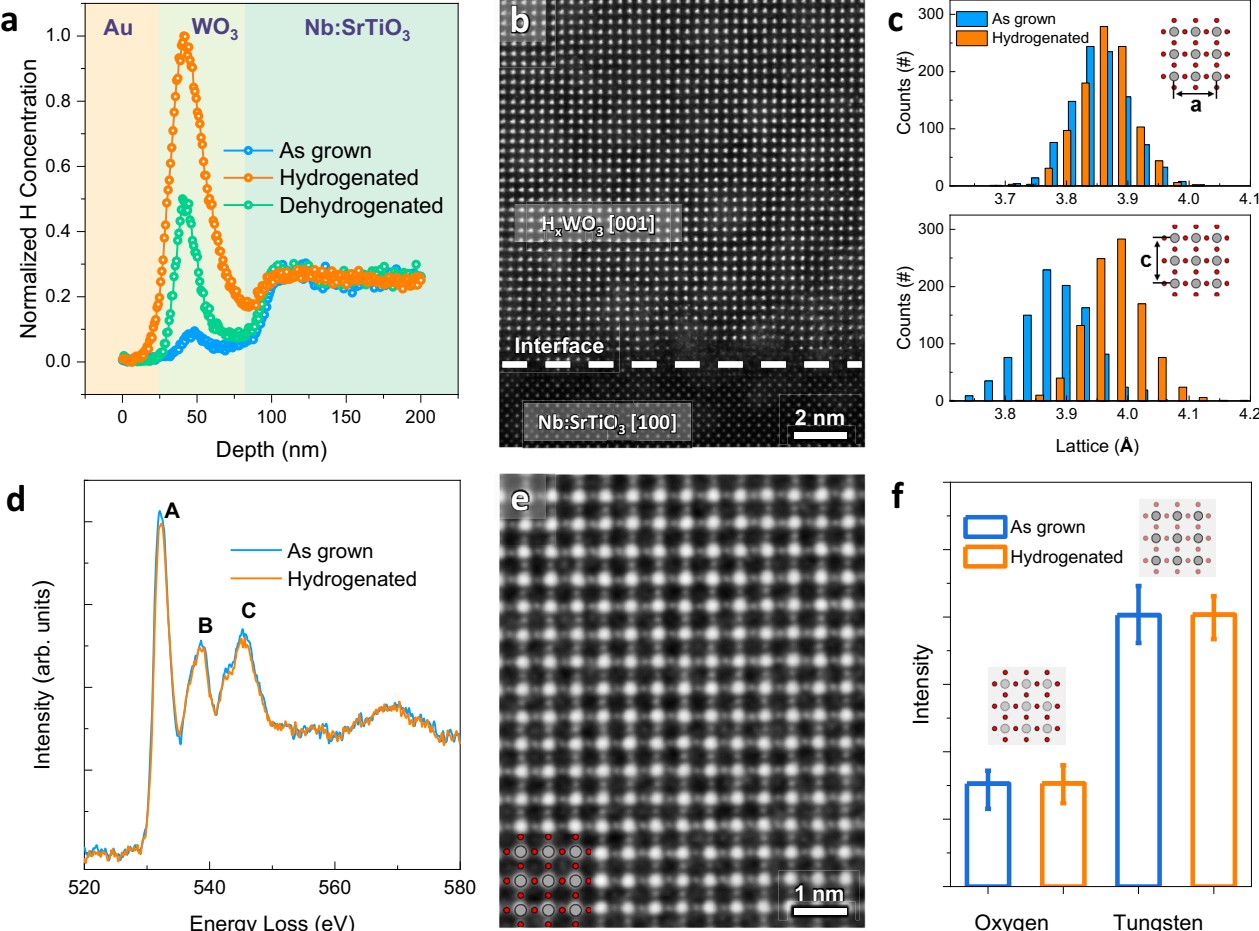

**Fig. 2 | Compositional and structural characterizations of the hydrogenated WO₃ samples. a** Depth profiles of hydrogen concentration for as-grown and hydrogenated WO₃ samples. The signals were normalized by the H intensity in the substrate. **b** High-angle annular dark-field (HAADF) image for the hydrogenated sample. **c** Statistical lattice constants before and after hydrogenation. **d** Electron energy loss spectra (EELS) at oxygen *K*-edges for as-grown and hydrogenated WO₃ samples. The labeled features in the spectra can be attributed to A: O2*p*-W*d*(t₂g), B: O2*p*-W*d*(e_g) and C: O2*p*-W6sp. **e** Integrated differential phase contrast (IDPC) image for hydrogenated sample. **f** Compositional analysis for W and O before and after the hydrogenation. The error bars shown here represent the standard deviation.

pronounced hydrogenation. Figure 3b summarizes the scan rate and time dependency of the conductivity for all pads demonstrated in Fig. 3a, which shows a critical switching speed of about 0.2 ms with a local current of ~4 nA. We note that this speed outreaches the hydrogenation of VO₂ samples (2 ms) by one order of magnitude[32], highlighting its potential for IMT switching devices. Interestingly, the current value is linearly proportional to the dwelling time during scanning (inset of Fig. 3b), providing a promising knob to precisely program the multiple resistive states. It could be expected that more investigations with pulsed voltage would be a useful way to further evaluate the intrinsic switching.

We subsequently probed the characteristic lateral scale for the probe-induced IMT by writing a series of conducting wires with the identical voltage of 3 V, as shown in Fig. 3c. Through the statistical analysis, the width of conducting wires is estimated to be about 230 nm. Further experiments reveal a correlation between the writing voltage and wire width, and a smaller width of about 75 nm was obtained at a lower voltage of 2 V (Supplementary Fig. 7). We note that this feature size is very close to the diameter of the scanning probe (~60 nm) employed, suggesting that the usage of a sharper probe might be able to further shrink the width. The lateral diffusion of hydrogen would be an inevitable effect during the tip-induced hydrogenation process, which can be enhanced by the stray field around the hemisphere-shaped tip/sample contact[53,54]. To better

control the hydrogenation resolution, which would benefit high-density data storage, further investigations need to be conducted to investigate electric-field-induced ionic (protonic) exchange at the tip/sample interface, where the charge injections[55], Joule heating[56], oxygen vacancies[57], and even strain field[58,59] could all play an essential role.

The close correlation between the local current and switching voltage suggests that the local current can be employed to encode a lithographic pattern with designed writing voltages. As a demonstration, we adapted a portrait of Albert Einstein to a lithographic template as shown in the bottom left corner of Fig. 3d, where the brightness of each pixel corresponds to a scanning voltage ranging between 0 to 3 V. Figure 3d demonstrates the obtained cAFM image from the scanned regions, and excitingly an undistorted photo is nicely obtained in which Einstein's hair and wrinkles can be clearly identified with the selected voltages (Supplementary Fig. 8). This experiment provides a direct showcase for the high-resolution nature of the probe-induced hydrogenation as well as its resulting multistate resistive switching.

To provide a quantitative investigation of the multistate resistive switching behaviors, we carried out successive writing and erasing processes to set the film into variant resistance states. The voltages of writing and erasing are identical in magnitude but opposite in sign. After each writing/erasing scan, a reading scan was conducted with cAFM to record the resistance state. Figure 4a illustrates the results obtained with a voltage of ±4 V, in which the sample gradually evolves

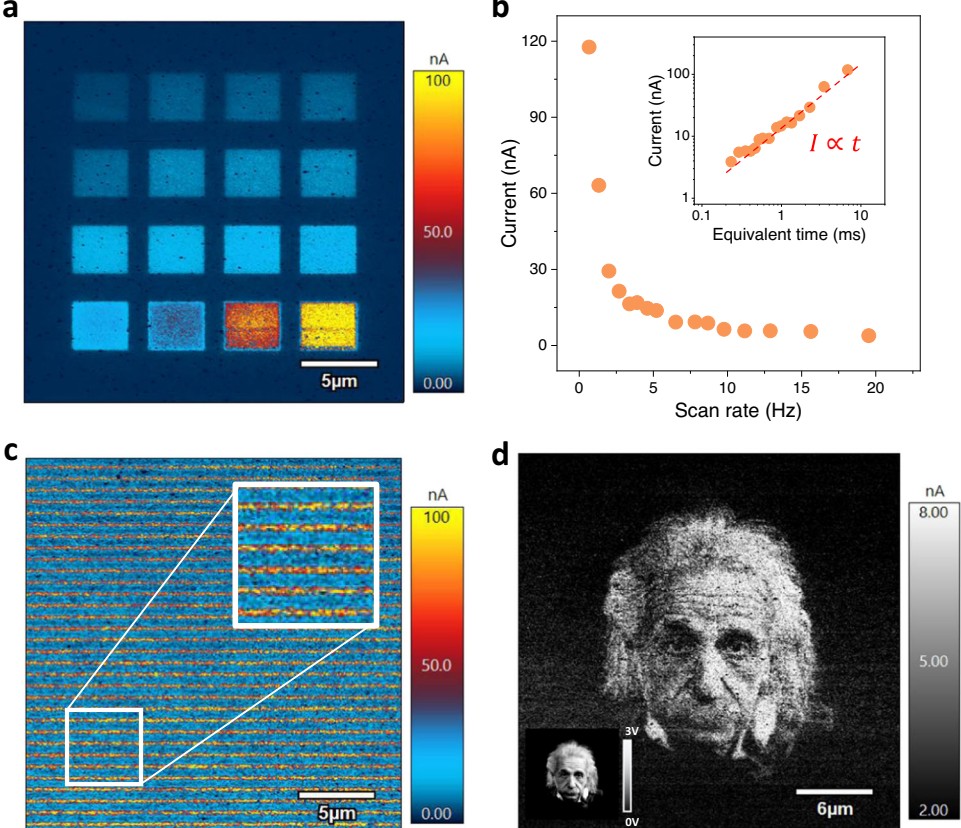

**Fig. 3 | Temporal and spatial modification of the probe-induced IMT within WO₃ thin films. a** Modulated local conductivity with different scanning speeds during the probe-induced hydrogenation. The 16 pads shown here are prepared with one time of writing scan with varied scan rates from 19.5 to 0.67 Hz (from the top left to the bottom right), while the written voltage was fixed at 3 V. **b** Correlation between the scan rate and the local conductivity illustrated in **a**. The inset curve gives the equivalent switching time dependence of the local conducting currents, in which the equivalent time was calculated by the equation of $\frac{P_0}{f_0} \times \frac{S_{tip}}{S_{scan}}$. $P_0$ is the number (128) of scan lines along the slow-scan direction, $f_0$ is the scan rate along the fast-scan direction, $S_{tip}$ is the estimated contact area (~400 nm²) between the probe and sample surface and $S_{scan}$ is the area of the pattern. **c** Formation of parallel arrays with 32 conducting wires. The lines were patterned with 3 V. **d** A portrait of Albert Einstein through probe-induced hydrogenation. (The original photo was taken by photographer Philipe Halsman; used with the permission of magnum/IC photo.) This image is lithographically patterned into WO₃ from a 0−3 V voltage template shown in the bottom left corner, and the image was encoded by the local current.

from an insulating (high-resistance) state to a metallic (low-resistance) state with the scans at +4 V, and then reverses back with the scans of −4 V. As a crucial metric requirement for device applications, the symmetric behavior is quantitatively shown in Fig. 4b, which also includes the data obtained at ±2 V and ±3 V (Supplementary Fig. 9). Interesting to note is that all these three normalized curves against the maximum value show almost identical dependence with the number of switching, suggesting a generic switching mechanism. We note that such a symmetric modulation effect is the most desirable feature for multistate resistive devices[1,2].

Reliability is another crucial metric for consideration in resistive switching. To prove the performance consistency, 200 consecutive I−V sweeps were conducted with a statically engaged probe. As shown in Fig. 4c, the I−V curves show a small fluctuation, but with persistent IMT characteristic features. The cycle-dependent switching capability is also demonstrated in Fig. 4d, which is the endurance measurements between different modulation windows. The voltages of ±4 V were used to set the resistance between a high-resistance state (HRS) and a low-resistance state (LSR), while ±2 V was used to set the resistance between the HRS and an intermediate-resistance state (IRS). Figure 4d shows that the characteristic resistive switching remains after 500 cycles without signatures of device failure, which further verifies the reversibility of the hydrogenation-induced IMT.

In summary, we provide compelling evidence to endorse the probe-induced hydrogenation in WO₃ as a reliable strategy to achieve multistate resistive switching. Inheriting the essential merit of high-resolution SPM technologies, the probe-induced hydrogenation approach demonstrates that the IMT can be controlled at nanoscales. Notably, this study further illustrates the generic compatibility of this probe-induced ionic evolution approach at room temperature, rendering a new pathway to manipulate the electronic states for hydrogenated materials at nanoscale.

## Methods

### Thin film fabrications

WO₃ thin films (~50 nm) were epitaxially grown on Nb: SrTiO₃ (001) (0.7% wt. doped) substrates through the pulsed laser deposition method. The deposition was conducted at a substrate temperature of 500 °C, and the O₂ partial pressure was controlled to be around 0.1 mbar. A pulsed laser with the wavelength of 248 nm was used to ablate a WO₃ ceramic target with a focused energy density of 0.5 J/cm² and a frequency of 2 Hz. After the deposition, samples were cooled down to room temperature in 5 mbar of O₂ with a cooling rate of 5 °C/min.

### Hydrogenation and conducting atomic force microscope

The probe-induced hydrogenation and conductivity mapping measurements were performed by a commercial SPM setup (Cypher ES, Oxford Instruments) equipped with temperature and gas environmental control accessories and Pt-coated conductive cantilevers

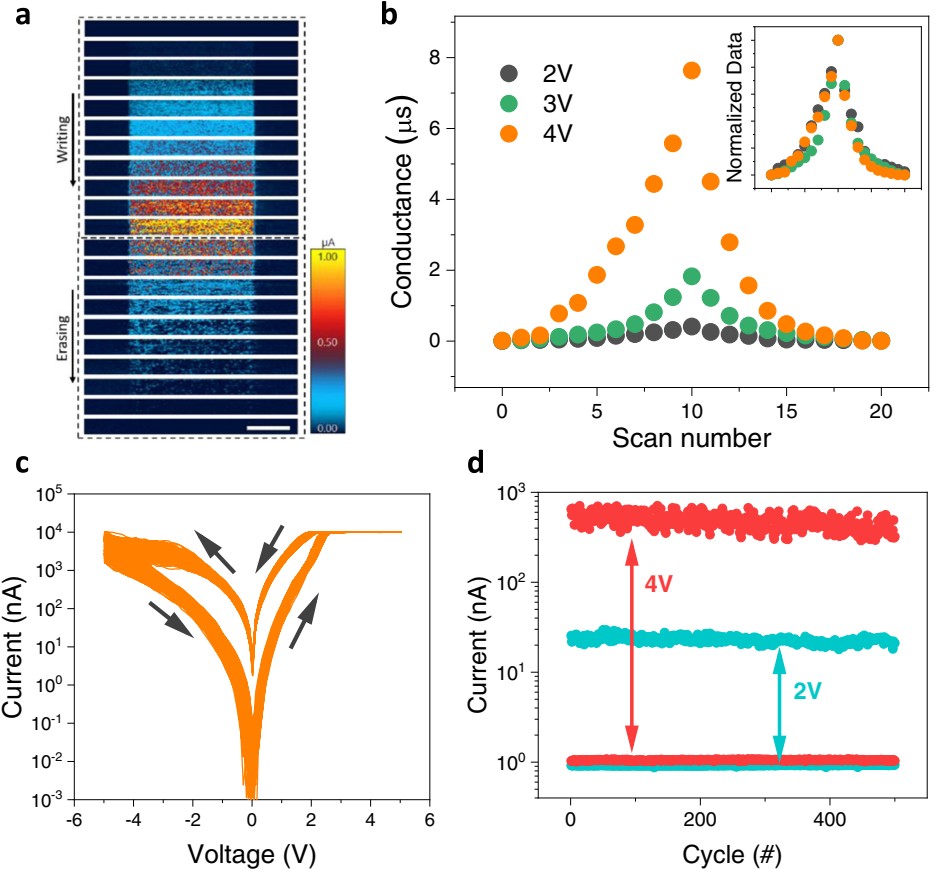

**Fig. 4 | Multistate resistive modulation through probe-induced hydrogen evolution. a** Maps of local current at an identical region through successive writing (hydrogenation) and erasing (dehydrogenation). The biased voltages of +4 V and −4 V at a scanning rate of 9.8 Hz were employed for the hydrogenation and dehydrogenation processes, respectively. The scale bar shows a 2 μm length. **b** A summary of conductance modulation through multiple successive writing and erasing processes. The inset shows the normalized conductance change among different voltages, in which the normalizations were performed against the maximum current value of each set of data. The nearly identical writing and erasing sequences for

these voltages nicely mimic the set/reset process for artificial synapse functions. The scan number indicates the number of scan times at the same area: 1–10 are successive writing scans and 11–20 are successive erasing scans. **c** Multiple (200) I–V curves obtained at a fixed point with the AFM probe as the top contact. The voltage sweeping rate is 20 V/s. **d** Endurance test of multiple-state resistive switching obtained at a fixed point. The biased voltages of +4 V/−4 V were used to set and reset between a high-resistance state (HRS) and a low-resistance state (LSR), as shown in the red curves; while +2 V/−2 V was employed to set the resistance between the HRS and an intermediate-resistance state (IRS).

---

(HQ:NSC18/Pt, MikroMasch, tip radius is ~30 nm). All the writing, erasing and reading experiments are conducted with an orca tip holder at room temperature. Under the orca module, the current measurement range is ±10 μA and the sensitivity is about 1 pA. All conducting atomic force microscope mapping measurements were conducted with a reading voltage of 0.1 V. To eliminate the influence of "water writing", the sample stage was first heated up to 120 °C with flowing argon gas for 30 min. After cooling down to room temperature, the SPM chamber was flushed by the forming gas (5% hydrogen/95% Ar) for 30 min. Finally, a continued forming gas flow was maintained during the hydrogenation process. The writing/erasing/reading voltage in this work describes the potential of the tip relative to the substrate.

### Scanning transmission electron microscope measurements

The cross-sectional scanning transmission electron microscope (STEM) specimen was prepared using the focused ion beam (FIB) instrument. The sample was thinned down using an accelerating voltage of 30 kV with a decreasing current from 240 to 50 pA, and then with a fine polishing process using an accelerating voltage of 5 kV and a current of 20 pA. The STEM-HAADF and STEM-IDPC images were acquired with an FEI Titan Cubed Themis 60–300 (operated at 300 kV), capable of recording high-resolution STEM images with a spatial resolution of ~0.059 nm. The microscope was equipped with a

high-brightness electron gun (X-FEG with monochromator), a spherical aberration corrector, and a post-column imaging energy filter (Gatan Quantum 965 Spectrometer). The energy resolution was smaller than 0.3 eV. The IDPC images were acquired with a four-segmented detector, and the collection angle was 5–27 mrad. The HAADF detector's collection angle was 48–200 mrad. The determination of the atomic position was obtained through the statSTEM[60]. The Dual-EELS mode was selected to collect core-loss signals of oxygen. The entrance aperture was 5 mm, and the step size was 0.1 eV. The EELS measurement was carried out using the line scan across the length of 1.2 μm, and the presented EELS data in Fig. 2d were the integrated signals through the scans at different regions. EELS data processing was carried out, including extraction of signals, calibration of the zero-loss position, pre-edge background subtraction, removal of multiple scattering, and normalization. All EELS data were processed using the Gatan Microscopy Suite 3.0 software package. The area for lattice parameter analysis is illustrated in Supplementary Fig. 6a, where the lattice parameters of pristine and hydrogenated regions are the averaged results of 225 (15 by 15) unit cells.

### Secondary ion mass spectroscopy measurements

SIMS measurements were carried out using a Tof-SIMS 5–10 instrument (IONTOF GmbH). The sputtering area was 350 × 350 μm, and the

detecting area was 150 ×150 μm in order to avoid disturbance from the crater edge. Due to the relatively small scanning range (30 μm) of our SPM setup, we patterned a 5 × 5 30 μm by 30 μm array to achieve the desired area for the SIMS measurements. Before the measurement, the whole sample was capped with Au through sputtering to reduce hydrogen release. The samples were prepared on conducting Nb-dopped $SrTiO_3$ substrates. During the SIMS measurements, the conducting substrate was grounded with aluminum foils to reduce the charge accumulation on the surface. The interface and surface positions were indexed through the measurements of Au, W, Sr, and Ti elements, while the H signal was simultaneously recorded. All samples were measured under nominally identical conditions, and the H signals were normalized by the background signal within the substrate.

## Data availability

All data supporting the results of this study are available in the manuscript or the supplementary information. Additional data are available from the corresponding author upon reasonable request.

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

## Acknowledgements

This study was financially supported by the National Key R&D Program of China (grant Nos. 2021YFE0107900 and 2021YFA1400300); the Basic Science Center Program of NSFC (grant No. 52388201); the Beijing Nature Science Foundation (grant No. Z200007); the National Natural Science Foundation of China (grants No. 52025024, 51872155, 12104250, and 52161135103). F.Z. and Y.Z. acknowledge support from the Shuimu Tsinghua Scholar Program of Tsinghua University. F.Z. also thanks the International Postdoctoral Exchange Fellowship Program of China (Talent-Introduction Program) (grant No. YJ20200328) for support. This work made use of the resources of the National Center for Electron Microscopy in Beijing.

## Author contributions

P.Y. conceived the project and designed the experiments. F.Z. grew the samples and performed the SPM-based hydrogenation, cAFM, XRD, SIMS, and data analysis with help from L.L., X.M., H.P., S.S., M.W., and K.X. Y.Z. carried out the TEM and EELS measurements and analyzed the data. X.M., S.J., D.Y., T.N., and J.T. provided useful insights for the characterization and understanding of the RS behaviors. F.Z., Y.Z., and P.Y. wrote the paper, while all authors discussed the results and commented on the manuscript.

## Competing interests

The authors declare no competing interests.
