## [Peer Review File · Nature Communications]

REVIEWER COMMENTS

Reviewer #1 (Remarks to the Author):

Zhang et.al. reported the nanoscale multistate resistive switching in WO₃ thin films through probe induced proton injection at room temperature. Oxygen vacancies based resistive switching has been investigated for long time (more than 20 years). However, oxygen vacancies in oxide materials cannot be quantitatively controlled like N type or P type dopants in semiconductors, leading to large cycle- to-cycle variation and cell-to-cell variation. Proton injection into oxide materials may be a good way to fabricate the controllable devices. In their previous work (Nat. Mater. 21, 1246 (2022)), a local and reversible electric-field-controlled insulator–metal transition through hydrogen evolution was demonstrated in VO₂ at elevated temperature, which is definitely not suitable for the application. In the current work, they have found WO₃ which has similar electric-field-controlled insulator–metal transition through hydrogen evolution, but at room temperature. In addition, nanoscale multistate resistive switching has been demonstrated by patterning Albert Einstein’s portrait with high resolution. The discovery of WO₃ and demonstration of nanoscale multistate switching make it deserve for publication in Nature Communication. However, I have some detailed comments which need the authors to address before publication.

1. Why the authors choose 50 nm WO₃? How the thickness affects the MI transition under the hydrogen evolution?
2. A typical I-V curve would be good for audience to understand the electric transport behaviour.
3. Authors use the SIMS to compare the hydrogen concentration between the as-grown sample and hydrogenated sample. It is known second ion ionization rate may depend on the electric states of the sample (insulating and conducting). Comments may be required in the manuscript.
4. How the multistate resistivity is related to hydrogen? The size of conducting path change or conducting carrier concentration change? What is expected proton gradient along thickness direction?

Reviewer #2 (Remarks to the Author):

This manuscript reports on resistive switching properties of WO₃ enabled through scanning probe hydrogenation that triggers an insulator-metal-transition. The use of Pt or Pd scanning microscopy

tips is shown to provide a catalytic behaviour in such process, allowing for localized, controllable conduction states throughout the area of the dielectric. The work is well organized and clearly written, though some grammar issues and typos can be found (some were pointed, but careful proofreading is encouraged). Some figures could benefit from small improvements. The methodology is consistent with that of other works on the topic, though some aspects of the characterization should be clarified for reproducibility and some figures of merit should be more carefully addressed (detailed in comments below).

The analysis and structure of the work largely resemble a recent study by the group (Ref. 30) for VO₂ on Al₂O₃ substrate. The main difference (apart from the change of materials to WO₃ on STO) is the data reported in Figure 4 demonstrating resistive switching behaviour. Although this could be interpreted as incremental contribution, the results are of potential interest and could be recommended for publication, but a few aspects may be worthy of revising before that stage. Find below some aspects that should be improved, from this reviewer's perspective (these comments are also attached as a .docx file for the authors' convenience).

1) In the introduction, scanning probe memories are mentioned in general, but with no reference provided about their importance or nature. Since it would be a niche for the reported phenomenon, some general description may aid on describing the potential of the technology.

2) STO is known for its great potential for forming atomically abrupt, conductive interfaces with other semiconductors and dielectrics, hence it has been explored in a wide variety of devices ranging from memristors [a,b] to photocatalysis cells [c]. However, in this paper, its role is not well described. This is a bit shocking since one of the main differences between this work and Ref. [30] is the replacement of Al₂O₃ as bottom electrode of the switching layer for STO. What is the reason for selecting highly doped STO in this case? In photocells, it aids on the provision of electrons to hydrogen ions. Does this characteristic or band alignment play any role here? A more detailed explanation could be helpful to strengthen the study.

[a] Boyeras Baldomá, S., Pazos, S. M., Aguirre, F. L., Ankonina, G., Kornblum, L., Yalon, E., & Palumbo, F. (2022). Wear-out and breakdown of Ta₂O₅/Nb:SrTiO₃ stacks. *Solid-State Electronics*, 198, 108462. <https://doi.org/10.1016/J.SSE.2022.108462>

[b] Miron, D., Cohen-Azarzar, D., Segev, N., Baskin, M., Palumbo, F., Yalon, E., & Kornblum, L. (2020). Band structure and electronic transport across Ta₂O₅/Nb:SrTiO₃ interfaces. *Journal of Applied Physics*, 128(4), 045306. <https://doi.org/10.1063/1.5139533>

[c] Kornblum, L., Fenning, D. P., Faucher, J., Hwang, J., Boni, A., Han, M. G., Morales-Acosta, M. D., Zhu, Y., Altman, E. I., Lee, M. L., Ahn, C. H., Walker, F. J., & Shao-Horn, Y. (2017). Solar hydrogen

production using epitaxial SrTiO₃ on a GaAs photovoltaic. *Energy & Environmental Science*, 10(1), 377–382. <https://doi.org/10.1039/C6EE03170F>

3) In supplementary Fig. 1b, a small correlation seems to be observed in the homogeneity of the map and the regions that were scanned for potentiation and depression. Can this be related to the additional sweep performed in the de-hydrogenated region, that cleans the surface a bit more?

4) In Supp. Fig. 2, the authors say: The writing voltage at the middle region was gradually increased from 0 V to 5 V. Does this mean that the voltage is increased as the scan takes place? Is that why the current profile shows higher currents on one side? If so, providing a voltage vs. position graph may help understand and replicate the experiment conditions.

5) A very important aspect of memristor reliability, specially for storage, is the retention time. The authors provide a discussion on the poor retention of oxygen driven switching in O₂ environments in Supp. Fig. 4. However, there is no mention to the retention of the reported phenomena, namely the one evaluated in a forming gas atmosphere. This is a fundamental analysis that should be provided, mainly after mentioning that the oxygen counterpart does not provide sufficiently good results.

6) Hydrogen catalytic capability is mentioned for Pd and Pt coated probes but it is not discussed nor referenced. Please, provide some details on the fundamentals behind this and/or some literature references to support this.

7) The “apparent suppression of peak A” in Fig. 2d is not too evident. What’s the uncertainty of each measurement? Could this be variability? Namely, if various (say, 10) measurements were performed, would this apparent suppression be consistent in all these? The figure in its current form does not give much supporting information to this specific claim.

This also applies to the data from the histograms of Fig. 2c. TEM images with high resolution focus on a very small cross section of the device, especially compared to the scanning tip area of influence (not smaller than 75 nm², as indicated further along the text). Therefore, I’d like to ask how large is the probed area that generates those histograms? Are these results consistent through various images? Also, can the authors provide a reference on the phenomenon driving this observed effect?

Perhaps this information could replace Fig. 2d which, from this reviewer’s perspective, does not provide much valuable information.

8) Please clarify in the main text (and maybe in the x-label of the inset of Fig. 3b) that the time is actually the cAFM reaction time, calculated as mentioned in the caption/methods. For the non-specialized reader in AFM, this would be helpful.

9) The claim on the enhanced speed of the hydrogenation process when compared to VO₂ samples it is interesting. However, in Ref. 30 this metric is inferred from the peaks in the Raman spectra after different experiments that used different scanning rates. In this work, however, the time is inferred from the measured change in conductivity (in-situ). So, are these two metrics comparable? I would expect both measurements extracted with the same technique for a fair comparison. At least, this should be mentioned in the text when performing such claim.

10) Voltage effects on the area at which hydrogenation-related resistance tuning is observed is a very interesting phenomenon. However, the authors do not suggest any origin for this nor provide systematic measurements showing its behaviour. What is this effect attributed to? Spreading of electric field lines? Lateral diffusivity of hydrogen within the WO₃? Since this is being showcased, at least a short comment could provide directions for further research.

11) Figure 3d, while impressive, shows peak currents that are much smaller (around 10 times) than in the rest of the experiments shown in the manuscript. Is there a reason for this? What was the scan rate in this experiment? Is there a correlation between scan rate and resolution (meaning by resolution the lateral dimension from panel c and Supp. Fig. 7)? Please clarify.

12) Fig. 4b shows in the inset its normalized data. Normalized against what magnitude?

13) Current in the loops of Fig. 4d is much higher (easily by 2 orders of magnitude) than the current reported on Fig. 1, even at lower applied voltages. What is the origin of such a large difference? What's the variability of the I-V loops in different regions of the material? If discussing reliability, as suggested by the authors in the next to last paragraph, this is important to be addressed.

14) In the same spirit as comment (12), I do not agree with saying that cycle-to-cycle variation is negligible in Fig. 4c. Rather than variability, and this is an impression without having the raw data, the observed effect seems more like a consistent drift with the accumulation of cycles, similar to the one observed in Fig. 4d for successive cycles. Therefore, this shows a cycle (and therefore, time) dependent degradation of the switching capability. I suggest avoiding the mention "negligible" and rather carefully address the observed drift with the accumulation of cycles/higher applied voltages (mimicking accelerated stress conditions).

15) At the end of next to last paragraph, the authors say they "demonstrate its good reliability". For 500 cycles of a single device, this claim is a bit too strong. It is a promising show of endurance, well

displayed for the scope of the work but, from this reviewer's perspective, it doesn't demonstrate "good reliability".

16) See typos and grammar, e.g.:

"To direct directly visualize the pattern, ..."

"The switching behaviors retains remains stable after 500 cycles ..."

"... in this work descripts describes the potential ..."

Reviewer #3 (Remarks to the Author):

In their work, the authors report an demonstration of a convenient scanning-probe-induced proton evolution within WO₃. In principle, the topic of the publication is appropriate for the Nature Communications; however, the paper needs to be revised with minor corrections before being accepted and the authors should take into account the following points:

Methods:

In this section, it should be included the information on the equipment used for RS properties measurements, the measurement ranges and the sensitivity of the equipment. The size of the contacts must also be placed to have a clear relation of the dimensionality of the WO₃ and the contacts for the I-V measurements.

We thank all reviewers for their constructive suggestions and comments, which are very valuable in improving the quality of our manuscript.

Reviewer #1 (Remarks to the Author):

Zhang et.al. reported the nanoscale multistate resistive switching in WO₃ thin films through probe induced proton injection at room temperature. Oxygen vacancies based resistive switching has been investigated for long time (more than 20 years). However, oxygen vacancies in oxide materials cannot be quantitatively controlled like N type or P type dopants in semiconductors, leading to large cycle- to-cycle variation and cell-to-cell variation. Proton injection into oxide materials may be a good way to fabricate the controllable devices. In their previous work (Nat. Mater. 21, 1246 (2022)), a local and reversible electric-field-controlled insulator–metal transition through hydrogen evolution was demonstrated in VO₂ at elevated temperature, which is definitely not suitable for the application. In the current work, they have found WO₃ which has similar electric-field-controlled insulator–metal transition through hydrogen evolution, but at room temperature. In addition, nanoscale multistate resistive switching has been demonstrated by patterning Albert Einstein’s portrait with high resolution. *The discovery of WO₃ and demonstration of nanoscale multistate switching make it deserve for publication in Nature Communication.* However, I have some detailed comments which need the authors to address before publication.

- 1) Why the authors choose 50 nm WO₃? How the thickness affects the MI transition under the hydrogen evolution?

RESPONSE: We thank the reviewer for this question. As requested by the reviewer, we have extended our studies with WO₃ thin films at different thickness (e.g., 10 nm, 50 nm, and 100 nm). As shown in **Fig. R1**, all samples are hydrogenated within a defined region through the scanning probe, in which a pronounced IMT is observed; while thinner samples have higher conductivity after hydrogen intercalation, since the conductance is inversely proportional to the sample thickness.

Figure R1. Conducting AFM maps of hydrogenated patterns on WO₃ samples with different thickness. The samples’ thicknesses are 10 nm, 50 nm and 100 nm, from left to the right. The hydrogenation voltage was set as +4 V.

2) A typical I-V curve would be good for audience to understand the electric transport behaviour.

RESPONSE: Thanks for the suggestion. **Fig. R2** shows I-V curves measured at the as-grown, the hydrogenated and the dehydrogenated regions. From the measurements, the hydrogenated region turns into low-resistance metallic state with characteristic linear I-V curve, while the as-grown and dehydrogenated areas remain at high-resistance insulating state. We included this figure in the revised supplementary information.

Figure R2. Characteristic I–V curves measured at different states. The black, red and blue lines are the I-V curves measured at the as-grown, the hydrogenated and the dehydrogenated regions, respectively.

3) Authors use the SIMS to compare the hydrogen concentration between the as-grown sample and hydrogenated sample. It is known second ion ionization rate may depend on the electric states of the sample (insulating and conducting). Comments may be required in the manuscript.

RESPONSE: We thank the reviewer for this suggestion. We note that the SIMS results provide qualitative evidence that the hydrogen ions were indeed intercalated into the WO_3 films using the scanning. Therefore, the distinct hydrogen profiles between the as-grown and hydrogenated samples (shown at **Fig. 2a** in the main text) directly confirm the existence of hydrogen in the hydrogenated sample with dramatically enhanced SIMS intensity. To reduce the electrostatic charge accumulation during the measurements, the samples are prepared on conducting substrate (i.e., Nb doped SrTiO_3), which is grounded during the measurement. This information was provided in the method section.

4) How the multistate resistivity is related to hydrogen? The size of conducting path

change or conducting carrier concentration change? What is expected proton gradient along thickness direction?

RESPONSE: We thank the reviewer again for this constructive comment. The multiple resistance states are attributed to the change of carrier concentration, which is related to the hydrogen content within WO_3 . We have previously studied the ionic liquid gating induced hydrogenation in WO_3 with the sheet resistance measured as different gating voltages¹, where the results reveal that the hydrogen concentration forms a direct/efficient tuning knob in modulating the sheet resistance of WO_3 thin films.

For conductive filaments induced memristive switching, the size of conducting path underneath the metal electrode is the most crucial factor determining the resistive state of the entire device. In this case, due to the relatively dilute distribution of the filaments, a high-resolution CAFM measurement would result in an inhomogeneous current map, as shown in a recent study of filament induced IMT². However, our current studies clearly reveal that the multi-resistance patterns manipulated by either hydrogenation voltage, time, or writing number (shown in **Supp. Fig. 2c, 3a** and **4a**, respectively) are very homogenous, which can then rule out the contribution of conducting nano filament.

The reviewer is correct that the proton gradient along the thickness direction would be an important factor to consider. Indeed, our SIMS result indicates that the hydrogen concentration is higher near the surface of the film, which might be related to the intrinsic ionic diffusion process, in which the top layer would have a higher hydrogen concentration.

We added the related discussion in the revised manuscript.

Reviewer #2 (Remarks to the Author):

This manuscript reports on resistive switching properties of WO_3 enabled through scanning probe hydrogenation that triggers an insulator-metal-transition. The use of Pt or Pd scanning microscopy tips is shown to provide a catalytic behaviour in such process, allowing for localized, controllable conduction states throughout the area of the dielectric. *The work is well organized and clearly written*, though some grammar issues and typos can be found (some were pointed, but careful proofreading is encouraged). Some figures could benefit from small improvements. The methodology is consistent with that of other works on the topic, though some aspects of the characterization should be clarified for reproducibility and some figures of merit should be more carefully addressed (detailed in comments below).

The analysis and structure of the work largely resemble a recent study by the group (Ref. 30) for VO_2 on Al_2O_3 substrate. The main difference (apart from the change of materials to WO_3 on STO) is the data reported in Figure 4 demonstrating resistive switching

behaviour. Although this could be interpreted as incremental contribution, *the results are of potential interest and could be recommended for publication*, but a few aspects may be worthy of revising before that stage. Find below some aspects that should be improved, from this reviewer's perspective (these comments are also attached as a .docx file for the authors' convenience).

- 1) **In the introduction, scanning probe memories are mentioned in general, but with no reference provided about their importance or nature. Since it would be a niche for the reported phenomenon, some general description may aid on describing the potential of the technology.**

RESPONSE: We thank the reviewer for this excellent suggestion. In the revised manuscript, we added the following discussions: *“The concept of scanning probe memories has drawn much research attention over the last few decades because of their intrinsic high-density storage capabilities and their unique ability to write and read in an efficient manner.”*^{3,4}

- 2) **STO is known for its great potential for forming atomically abrupt, conductive interfaces with other semiconductors and dielectrics, hence it has been explored in a wide variety of devices ranging from memristors [a,b] to photocatalysis cells [c]. However, in this paper, its role is not well described. This is a bit shocking since one of the main differences between this work and Ref. [30] is the replacement of Al₂O₃ as bottom electrode of the switching layer for STO. What is the reason for selecting highly doped STO in this case? In photocells, it aids on the provision of electrons to hydrogen ions. Does this characteristic or band alignment play any role here? A more detailed explanation could be helpful to strengthen the study.**

RESPONSE: We appreciate the reviewer to raising this comment. In this study, we chose the highly Nb-doped STO (0.7% wt. Nb) to facilitate the conducting-AFM experiments. As reported in Ref. [30] (Ref. [32] in the revised manuscript), Au side electrodes are used on VO₂/Al₂O₃ samples, where the conducting-AFM can only be performed near the edge of the electrodes. Clearly, a heterostructure with a conducting substrate as the bottom electrode would simplify the device architecture, where the applied electric field can directly manipulate the hydrogenation level of the film⁵.

We note that the hydrogenation-induced IMT in WO₃ has been studied by ionic liquid gating and solid-state proton electrolyte^{1,6-8}. Among those studies, either LaAlO₃, YAlO₃ or SiO₂ is involved as the substrate to fabricate the WO₃ films, and similar resistance switching behaviors are observed. Therefore, the hydrogenation and the associated electron doping effect play a dominant role in the resistance switching. However, we agree with the reviewer that band alignment engineering would be an effective path to further optimize the performance of resistance switching, which forms an important and interesting project for future studies. To

address the roles of NSTO, the following text was added in the revised manuscript “*In this study, we employed the highly doped Nb:STO substrates⁹⁻¹³ as a bottom electrode to facilitate the tip induced hydrogenation as well as subsequent cAFM measurements.*”

3) In supplementary Fig. 1b, a small correlation seems to be observed in the homogeneity of the map and the regions that were scanned for potentiation and depression. Can this be related to the additional sweep performed in the dehydrogenated region, that cleans the surface a bit more?

RESPONSE: We note that the slight change in topography (**Fig. R3**) should be attributed to the hydrogenation-induced lattice expansion. This result is consistent with the results obtained from the TEM analysis, where notable lattice expansion (2.3%) along the c-axis is observed at the hydrogenated region. We included this discussion in the revised supplementary information.

Figure R3. Topography measurements around the hydrogenated and dehydrogenated regions. a, Topography obtained at a region with the as-grown, the hydrogenated and the dehydrogenated states. These different states are labeled with white dot lines. **b,** Topography profile crossing three different regions, as labeled by the blue dash line in (a).

4) In Supp. Fig.2, the authors say: The writing voltage at the middle region was gradually increased from 0 V to 5 V. Does this mean that the voltage is increased as the scan takes place? Is that why the current profile shows higher currents on one side? If so, providing a voltage vs. position graph may help understand and replicate the experiment conditions.

RESPONSE: The reviewer is correct that the voltage gradually increases during the scanning, as illustrated in **Fig. R4d**, in which the voltage increases linearly from the left to the right. Similarly, a voltage template (with negatively biased voltages) was employed for the dehydrogenation process, after which a current map was obtained, as shown in **Fig. R4f**. We have updated the **Supp. Fig. 2** with **Fig. R4**.

Figure R4. Probe-induced IMT at various gas environments. **a-c**, Probe-induced local conductivity modulation carried out in **(a)** forming gas (5% H₂ + 95% Ar), **(b)** pure O₂, and **(c)** pure Ar. The writing voltage at the middle region gradually increased from 0 V to 5 V. Data shown here are summarized and plotted in **Fig. 1c** as current profiles. Compared with the pronounced conductivity observed in **(a)** due to the hydrogenation effect, the measurements at pure O₂ and pure Ar show much-reduced conductivity. **d**, Biased voltage vs. position used during the writing process in **(a-c)**. **e**, Greyscale voltage template used for the lithographic hydrogenation in **(a-c)**. **f**, Voltage-dependent dehydrogenation process from a hydrogenated sample. The application of negative biased voltages dehydrogenates the sample with the recovery of an insulating state. All conducting maps shown here employ the same color scale bar as shown on the right.

5) A very important aspect of memristor reliability, specially for storage, is the retention time. The authors provide a discussion on the poor retention of oxygen driven switching in O₂ environments in Supp. Fig. 4. However, there is no mention to the retention of the reported phenomena, namely the one evaluated in a forming gas atmosphere. This is a fundamental analysis that should be provided, mainly after mentioning that the oxygen counterpart does not provide sufficiently good results.

RESPONSE: As suggested by the reviewer, the retention test was performed in the forming gas environment as well, in which a conducting region was continuously measured with the results shown in **Fig. R5**. It shows that the hydrogenated area sustains a high conductivity state up to 300 minutes after the hydrogen intercalated. At the same time, a notable change in the conducting region was also observed in the horizontal axis, which should be attributed to the tip-induced protonic out-diffusion from the central region. Nevertheless, the hydrogenated sample still shows notably longer retention as compared with the case for the sample with

oxygen vacancies, which has a retention time of a few minutes with strongly reduced current. This information is also included in the revised supplementary information.

Figure R5. Retention test of the tip-induced hydrogenation within forming gas.

- 6) Hydrogen catalytic capability is mentioned for Pd and Pt coated probes but it is not discussed not referenced. Please, provide some details on the fundamentals behind this and/or some literature references to support this.

RESPONSE: As suggested, we added the following discussion in the revised manuscript “We note that noble metals of Pt and Pd are the most widely employed catalysts used for hydrogen spillover.¹⁴ However, compared with Pd, Pt has smaller activation energy for the catalytic reaction, meaning it can facilitate hydrogen spillover in a more efficient manner¹⁵, leading to a more pronounced conducting state with identical writing conditions.”

- 7) The “apparent suppression of peak A” in Fig. 2d is not too evident. What’s the uncertainty of each measurement? Could this be variability? Namely, if various (say, 10) measurements were performed, would this apparent suppression be consistent in all these? The figure in its current form does not give much supporting information

to this specific claim.

This also applies to the data from the histograms of Fig. 2c. TEM images with high resolution focus on a very small cross section of the device, especially compared to the scanning tip area of influence (not smaller than 75 nm², as indicated further along the text). Therefore, I'd like to ask how large is the probed area that generates those histograms? Are these results consistent through various images? Also, can the authors provide a reference on the phenomenon driving this observed effect?

Perhaps this information could replace Fig. 2d which, from this reviewer's perspective, does not provide much valuable information.

RESPONSE: We thank the reviewer for this critical comment regarding the STEM measurements. We note that the suppression of peak A is attributed to the electron doping effect associated with hydrogenation. Specifically, the EELS measurement was carried out using the line scan across the length of 1.2 μm (**Fig. R6 left**), and the presented EELS data in **Fig. 2d** were the integrated signals through the scans at different regions. To further verify the trend, we also extracted signals from a smaller window of 0.3 μm integrated width (**Fig. R6 right**), from which almost identical features were obtained.

Regarding the quantitative measurements of lattice constants shown in **Fig. 2c**, we showed the analyzed area in the **Supp. Fig. 6a**, where the lattice parameters of pristine and hydrogenated regions are the averaged results of 225 (15 by 15) unit cells. It is important to note that, through this analysis, a notable lattice expansion along the out-of-plane direction was observed, which is consistent with the scenario of hydrogenation induced lattice expansion as observed in previous studies^{6, 16-18}; for instance, our previous study¹ of ionic liquid gating induced hydrogenation into WO_3 with emergent IMT shows a chemical expansion up to 3.5%,

We have added the related discussion in the revised manuscript.

Figure R6. EELS measurements around a larger area. Left panel: line scan regions for the EELS measurements. **Right panel:** Representative O K-edge EELS spectra taken at pristine and hydrogenated regions. The integrated width for data extraction is 0.3 μm .

8) Please clarify in the main text (and maybe in the x-label of the inset of Fig. 3b) that the time is actually the cAFM reaction time, calculated as mentioned in the caption/methods. For the non-specialized reader in AFM, this would be helpful.

RESPONSE: Thanks for the suggestion. As suggested, we revised the manuscript and figure caption as suggested, and specifically, the time is now changed to equivalent time, as it is a calculated value based on other measurement parameters as discussed in the methods.

9) The claim on the enhanced speed of the hydrogenation process when compared to VO₂ samples it is interesting. However, in Ref. 30 this metric is inferred from the peaks in the Raman spectra after different experiments that used different scanning rates. In this work, however, the time is inferred from the measured change in conductivity (in-situ). So, are these two metrics comparable? I would expect both measurements extracted with the same technique for a fair comparison. At least, this should be mentioned in the text when performing such claim.

RESPONSE: We note that both VO₂ and WO₃ employed in the previous and current studies demonstrate an IMT through hydrogenation induced electron doping. Importantly, the IMT in

VO₂ is also accompanied with a structural transition, where the V–V dimerization is suppressed, and this feature is employed as an indicator to trace the phase transition through Raman spectra¹⁹. We note that the lack of bottom electron hinders and complicates the electrical measurement of VO₂ across the IMT, while the Raman spectra do not require electrical contact. For the WO₃ sample, the electron doping into empty 5d orbitals results in a metallic state, through which the perovskite structure remains. Therefore, the Raman is not an appropriate method to trace this transition. While, luckily the WO₃ was grown successfully on a conducting electrode (Nb:STO), in which a direct electrical measurement is feasible. Although different methods have been employed to trace the IMT phase transition of these two systems, the phase transition itself holds a characteristic feature of the materials, which is highly correlated with the intrinsic bulk proton diffusion coefficients, surface facets, etc. Through these studies, although the hydrogenation was carried out at elevated temperature (50 °C) and higher voltage (10 V) for VO₂, its phase transition speed (~2 ms) is still about one order of magnitude longer than that for the WO₃, which was carried out at room temperature with a smaller voltage of 3 V. This comparison clearly highlights the promising potential of WO₃ for IMT switching devices through proton evolution.

10) Voltage effects on the area at which hydrogenation-related resistance tuning is observed is a very interesting phenomenon. However, the authors do not suggest any origin for this nor provide systematic measurements showing its behaviour. What is this effect attributed to? Spreading of electric field lines? Lateral diffusivity of hydrogen within the WO₃? Since this is being showcased, at least a short comment could provide directions for further research.

RESPONSE: The reviewer brought out a very nice point here. Indeed, this should be attributed to the stray field induced lateral diffusion. From the geometric feature of the nano-sized scanning probe used, an inhomogeneous electric field (stray field) would form around the hemisphere-shape tip/sample contact regions^{20,21}, leading to lateral diffusion of the proton ions. An intuitive picture is that, with a higher voltage applied between the tip and the bottom electrode, the spreading electric field would be larger, and stronger lateral diffusion would be induced. However, a microscopic picture behind this would involve several interesting (though complicated) interactions, such as tip induced charge injections²², Joule heating²³, oxygen vacancy²⁴ and strain field^{25, 26}, and these factors would couple with the electric field to manipulate the observed phenomena of tip-induced hydrogenation. So, as suggested by the reviewer, we added the following comment about the voltage effect: *“The lateral diffusion of hydrogen would be an inevitable effect during the tip-induced hydrogenation process, which can be further manipulated through the stray field around the hemisphere-shape tip/sample contact.”*^{20,21} *To better control the hydrogenation resolution, which would benefit high-density data storage, further investigations need to be conducted to investigate electric field induced ionic (protonic) exchange at the tip/sample interface, where the charge injections²², Joule*

heating²³, oxygen vacancy²⁴ and even strain field^{25, 26} could all play an essential role.”

11) Figure 3d, while impressive, shows peak currents that are much smaller (around 10 times) than in the rest of the experiments shown in the manuscript. Is there a reason for this? What was the scan rate in this experiment? Is there a correlation between scan rate and resolution (meaning by resolution the lateral dimension from panel c and Supp. Fig. 7)? Please clarify.

RESPONSE: We thank the reviewer for the question. The writing of **Fig. 3d** was performed at the scanning rate of 1 Hz and with the voltage template of 0 to 3 V. We note that both the scanning rate and tip voltage during writing play an important role in the observed electric current. In the voltage lithographic template used for **Fig. 3d**, most of the pixels have a voltage smaller than 3 V (the maximum value). For instance, the voltage used to sketch the hair and face is about 0.8-2.7 V (The writing voltage at the red and blue circle in the **Fig. R7** is about 0.87 V and 2.73 V, respectively). We note that this voltage difference is the main reason resulting in the different currents, as there is a close correlation between them, as shown in **Fig. 1c**. While the resolution is mainly determined by the external voltage, and for instance, the voltage used for **Fig. 3c** is 3 V, which results in the lateral resolution of 230 nm, while a slightly reduced biased voltage of 2 V (**Supp. Fig. 7**) leads to the characteristic width of 75 nm for the conducting wires. We note that this trend can be attributed to the stray field induced proton diffusion along the lateral direction during the writing, in which the larger voltage means a stronger stray field, leading to the transport of protons at a longer distance. We have revised the manuscript to clarify these issues.

Figure R7. The portrait of Albert Einstein through probe-induced hydrogenation. The writing voltages at Einstein's hair in the red circle and nose in the blue circle are about 0.87 V and 2.73 V, respectively.

12) Fig. 4b shows in the inset its normalized data. Normalized against what magnitude?

RESPONSE: We thank the reviewer for this suggestion. These data were normalized with the maximum current to illustrate the trend through switching better. We have clarified this point in the figure caption: *“The inset shows the normalized conductance change among different voltages, in which the normalizations were performed against the maximum current value.”*

13) Current in the loops of Fig. 4d is much higher (easily by 2 orders of magnitude) than the current reported on Fig. 1, even at lower applied voltages. What is the origin of such a large difference? What's the variability of the I-V loops in different regions of the material? If discussing reliability, as suggested by the authors in the next to last paragraph, this is important to be addressed.

RESPONSE: We thank the reviewer for this critical reading and comparison. The data shown in **Fig. 4d** were measured through the single point electrical measurement with a static tip, in which the current is ~ 500 nA at the low resistance state set by 4 V and about 20 nA at the intermedia state set by 2 V. While the data in **Fig. 1c** were patterned and measured by a scanning

tip, in which the current is about 220 nA for 4V and about 15 nA for 2 V. We note this difference should be attributed to the different time duration during the writing and possibly the contact area between the sample and tip as well.²⁷⁻³⁰ As suggested by the reviewers, we have also performed multiple I-V loops at several randomly selected regions on the same sample (**Fig. R8**), in which the resistive switching phenomena were consistently observed. We have included this information in the revised supplementary information.

Figure R8. I-V measurement at 5 randomly chosen spots A-E on the same sample. The last figure is an overlay of these loops.

14) In the same spirit as comment (12), I do not agree with saying that cycle-to-cycle variation is negligible in Fig. 4c. Rather than variability, and this is an impression without having the raw data, the observed effect seems more like a consistent drift with the accumulation of cycles, similar to the one observed in Fig. 4d for successive cycles. Therefore, this shows a cycle (and therefore, time) dependent degradation of the switching capability. I suggest avoiding the mention “negligible” and rather carefully address the observed drift with the accumulation of cycles/higher applied voltages (mimicking accelerated stress conditions).

RESPONSE: We thank the reviewer for this critical comment. As suggested, we described the cycle-to-cycle variation as “*the I-V curves show a small fluctuation, but with persistent IMT characteristic features.*” We also mentioned the small accumulated drift observed in **Fig. 4d** in the revised manuscript: “*The characteristic resistive switching remains after 500 cycles with a small accumulated drift, which further verifies the reversibility of the hydrogenation induced IMT.*”.

15) At the end of next to last paragraph, the authors say they “demonstrate its good reliability”. For 500 cycles of a single device, this claim is a bit too strong. It is a promising show of endurance, well displayed for the scope of the work but, from this

reviewer's perspective, it doesn't demonstrate "good reliability".

RESPONSE: We thank the reviewer for this comment. In this revised manuscript, we changed this statement as discussed above.

16) See typos and grammar, e.g.:

"To direct directly visualize the pattern, ..."

"The switching behaviors retains remains stable after 500 cycles ..."

"... in this work descripts describes the potential ..."

RESPONSE: We thank the reviewer for the careful review. In the revised submission, we corrected those mistakes and further polished the writing.

Reviewer #3 (Remarks to the Author):

In their work, the authors report a demonstration of a convenient scanning-probe-induced proton evolution within WO₃. *In principle, the topic of the publication is appropriate for the Nature Communications*; however, the paper needs to be revised with minor corrections before being accepted and the authors should take into account the following points:

Methods:

In this section, it should be included the information on the equipment used for RS properties measurements, the measurement ranges and the sensitivity of the equipment. The size of the contacts must also be placed to have a clear relation of the dimensionality of the WO₃ and the contacts for the I-V measurements.

RESPONSE: We thank the reviewer for these comments. The electrical measurements were carried out with an SPM setup (model Cypher ES) from Oxford Instruments, which is equipped with an ORCA module for the current measurement. With this setup, the current range is ± 10 μ A and the sensitivity is ~ 1 pA. To perform the measurements, commercial Pt-coated conductive tips (model HQ:NSC18/Pt) from MikroMasch were employed. **Figure R9** is an SEM image of a typical HQ:NSC18/Pt tip obtained from the vender's website, from which the radius of the tip is estimated to be about 20~30 nm. We note that this value is notably smaller than the characteristic length scale (~ 75 nm) of the conducting wires obtained in this study, suggesting that the proton diffusion along the lateral direction forms an essential constraint for the experimentally obtained feature size in the current study. We added this information in the revised manuscript.

Figure R9. Scanning electron microscopy image of a typical HQ:NSC18/Pt tip from **MikroMasch**. This picture is downloaded from the vender's website.

Reference:

1. Wang, M.; Shen, S.; Ni, J.; Lu, N.; Li, Z.; Li, H.-B.; Yang, S.; Chen, T.; Guo, J.; Wang, Y.; et al. Electric-Field-Controlled Phase Transformation in WO₃ Thin Films through Hydrogen Evolution. *Advanced Materials* **2017**, *29* (46), 1703628.
2. Wei, T.; Lu, Y.; Zhang, F.; Tang, J.; Gao, B.; Yu, P.; Qian, H.; Wu, H. Three-Dimensional Reconstruction of Conductive Filaments in HfO_x-Based Memristor. *Advanced Materials* **2023**, *35* (10), 2209925.
3. Wang, L.; Yang, C. H.; Wen, J.; Gong, S. D.; Peng, Y. X. Overview of Probe-based Storage Technologies. *Nanoscale Research Letters* **2016**, *11* (1), 342.
4. Lee, M. H.; Hwang, C. S. Resistive switching memory: observations with scanning probe microscopy. *Nanoscale* **2011**, *3* (2), 490-502.
5. Li, H.-B.; Lou, F.; Wang, Y.; Zhang, Y.; Zhang, Q.; Wu, D.; Li, Z.; Wang, M.; Huang, T.; Lyu, Y.; et al. Electric Field-Controlled Multistep Proton Evolution in H_xSrCoO_{2.5} with Formation of H-H Dimer. *Advanced Science* **2019**, *6* (20), 1901432.
6. Leng, X.; Pereiro, J.; Strle, J.; Dubuis, G.; Bollinger, A. T.; Gozar, A.; Wu, J.; Litombe, N.; Panagopoulos, C.; Pavuna, D.; et al. Insulator to metal transition in WO₃ induced by electrolyte gating. *npj Quantum Materials* **2017**, *2* (1), 35.
7. Yang, J.-T.; Ge, C.; Du, J.-Y.; Huang, H.-Y.; He, M.; Wang, C.; Lu, H.-B.; Yang, G.-Z.; Jin, K.-J. Artificial Synapses Emulated by an Electrolyte-Gated Tungsten-Oxide Transistor. *Advanced Materials* **2018**, *30* (34), 1801548.
8. Yao, X.; Klyukin, K.; Lu, W.; Onen, M.; Ryu, S.; Kim, D.; Emond, N.; Waluyo, I.; Hunt, A.; del Alamo, J. A.; et al. Protonic solid-state electrochemical synapse for physical neural networks. *Nature Communications* **2020**, *11* (1), 3134.
9. Miron, D.; Cohen-Azarzar, D.; Segev, N.; Baskin, M.; Palumbo, F.; Yalon, E.; Kornblum, L.

Band structure and electronic transport across Ta₂O₅/Nb:SrTiO₃ interfaces. *Journal of Applied Physics* **2020**, *128* (4), 045306.

10. Zhang, Q.; Li, N.; Zhang, T.; Dong, D.; Yang, Y.; Wang, Y.; Dong, Z.; Shen, J.; Zhou, T.; Liang, Y.; et al. Enhanced gain and detectivity of unipolar barrier solar blind avalanche photodetector via lattice and band engineering. *Nature Communications* **2023**, *14* (1), 418.

11. Wen, Z.; Li, C.; Wu, D.; Li, A.; Ming, N. Ferroelectric-field-effect-enhanced electroresistance in metal/ferroelectric/semiconductor tunnel junctions. *Nature Materials* **2013**, *12* (7), 617-621.

12. Waser, R.; Aono, M. Nanoionics-based resistive switching memories. *Nature Materials* **2007**, *6* (11), 833-840.

13. Dittmann, R.; Menzel, S.; Waser, R. Nanoionic memristive phenomena in metal oxides: the valence change mechanism. *Advances in Physics* **2021**, *70* (2), 155-349.

14. Prins, R. Hydrogen Spillover. Facts and Fiction. *Chemical Reviews* **2012**, *112* (5), 2714-2738.

15. Yamauchi, M.; Kobayashi, H.; Kitagawa, H. Hydrogen Storage Mediated by Pd and Pt Nanoparticles. *ChemPhysChem* **2009**, *10* (15), 2566-2576.

16. Jarman, R.; Dickens, P. Electrochemical Insertion of Hydrogen in WO₃. *Journal of The Electrochemical Society* **1982**, *129* (10), 2276.

17. Xi, Y.; Zhang, Q.; Cheng, H. Mechanism of Hydrogen Spillover on WO₃ (001) and Formation of H_xWO₃ (x= 0.125, 0.25, 0.375, and 0.5). *The Journal of Physical Chemistry C* **2014**, *118* (1), 494-501.

18. Manca, N.; Mattoni, G.; Pelassa, M.; Venstra, W. J.; van der Zant, H. S. J.; Caviglia, A. D. Large Tunability of Strain in WO₃ Single-Crystal Microresonators Controlled by Exposure to H₂ Gas. *ACS Applied Materials & Interfaces* **2019**, *11* (47), 44438-44443.

19. Li, L.; Wang, M.; Zhou, Y.; Zhang, Y.; Zhang, F.; Wu, Y.; Wang, Y.; Lyu, Y.; Lu, N.; Wang, G.; et al. Manipulating the insulator–metal transition through tip-induced hydrogenation. *Nature Materials* **2022**, *21*(11), 1246-1251.

20. Hui, F.; Lanza, M. Scanning probe microscopy for advanced nanoelectronics. *Nature Electronics* **2019**, *2* (6), 221-229.

21. Strelcov, E.; Yang, S. M.; Jesse, S.; Balke, N.; Vasudevan, R. K.; Kalinin, S. V. Solid-state electrochemistry on the nanometer and atomic scales: the scanning probe microscopy approach. *Nanoscale* **2016**, *8* (29), 13838-13858.

22. Balke, N.; Maksymovych, P.; Jesse, S.; Kravchenko, I. I.; Li, Q.; Kalinin, S. V. Exploring Local Electrostatic Effects with Scanning Probe Microscopy: Implications for Piezoresponse Force Microscopy and Triboelectricity. *ACS Nano* **2014**, *8* (10), 10229-10236.

23. Kim, Y.; Kumar, A.; Tselev, A.; Kravchenko, I. I.; Han, H.; Vrejoiu, I.; Lee, W.; Hesse, D.; Alexe, M.; Kalinin, S. V.; et al. Nonlinear Phenomena in Multiferroic Nanocapacitors: Joule Heating and Electromechanical Effects. *ACS Nano* **2011**, *5* (11), 9104-9112.

24. Das, S.; Wang, B.; Cao, Y.; Rae Cho, M.; Jae Shin, Y.; Mo Yang, S.; Wang, L.; Kim, M.;

- Kalinin, S. V.; Chen, L.-Q.; et al. Controlled manipulation of oxygen vacancies using nanoscale flexoelectricity. *Nature Communications* **2017**, *8* (1), 615.
25. Park, S. M.; Wang, B.; Das, S.; Chae, S. C.; Chung, J.-S.; Yoon, J.-G.; Chen, L.-Q.; Yang, S. M.; Noh, T. W. Selective control of multiple ferroelectric switching pathways using a trailing flexoelectric field. *Nature Nanotechnology* **2018**, *13* (5), 366-370.
26. Lu, H.; Bark, C.-W.; Esque de los Ojos, D.; Alcalá, J.; Eom, C. B.; Catalan, G.; Gruverman, A. Mechanical Writing of Ferroelectric Polarization. *Science* **2012**, *336* (6077), 59-61.
27. Mizzi, C. A.; Lin, A. Y. W.; Marks, L. D. Does Flexoelectricity Drive Triboelectricity? *Physical Review Letters* **2019**, *123* (11), 116103.
28. Xu, G.; Hao, F.; Weng, M.; Hong, J.; Pan, F.; Fang, D. Strong influence of strain gradient on lithium diffusion: flexo-diffusion effect. *Nanoscale* **2020**, *12* (28), 15175-15184.
29. Enachescu, M.; van den Oetelaar, R. J. A.; Carpick, R. W.; Ogletree, D. F.; Flipse, C. F. J.; Salmeron, M. Atomic Force Microscopy Study of an Ideally Hard Contact: The Diamond(111)/Tungsten Carbide Interface. *Physical Review Letters* **1998**, *81* (9), 1877-1880.
30. Celano, U.; Hantschel, T.; Giammaria, G.; Chintala, R. C.; Conard, T.; Bender, H.; Vandervorst, W. Evaluation of the electrical contact area in contact-mode scanning probe microscopy. *Journal of Applied Physics* **2015**, *117* (21), 214305.

REVIEWERS' COMMENTS

Reviewer #1 (Remarks to the Author):

The authors have addressed my concerns in their response. I recommend it for publication.

Reviewer #2 (Remarks to the Author):

I'd like to thank the authors for taking the time to address my comments. They have provided additional data, both in the main text and the supplementary information, that greatly enhances the overall understanding and reach of their work and its reproducibility.

I'm happy to recommend this work for publication. I would only suggest the following points to be included in the final version:

1) It would help a lot for future research to clearly indicate if the device that was measured for 500 cycles stopped working or the measurement was just stopped due to time extension. If it failed, address why it failed (stopped switching, got stuck at one resistance level, shorted out, the tip degraded). This information can be very helpful for further exploring the actual reliability of the phenomenon.

2) Please, whenever reporting pinched hysteron I-V curves in resistive switching devices, include the ramp rate in V/s either as a label in the plot or within the caption. This information is crucial to have a grasp into the dynamics of the phenomenon instead of understanding it as a completely stationary process, which is never the case in resistive switching devices.

3) For future research, addressing the switching phenomenon by applying pulsed voltage well below 1 ms should be fundamental to understand the true potential of the technology in memory/storage applications.

Reviewer #3 (Remarks to the Author):

In their work, the authors report an demonstration of a convenient scanning-probe-induced proton evolution within WO₃.

In principle, the topic of the publication is appropriate for the Nature Communications;

The authors welcomed all comments and the article is more complete, therefore, the article is accepted for publication.

1) It would help a lot for future research to clearly indicate if the device that was measured for 500 cycles stopped working or the measurement was just stopped due to time extension. If it failed, address why it failed (stopped switching, got stucked at one resistance level, shorted out, the tip degraded). This information can be very helpful for further exploring the actual reliability of the phenomenon.

RESPONSE: The measurement is set for 500 cycles and stopped without signatures of device failure. We added this information in the revised manuscript when discussing the relevant results.

2) Please, whenever reporting pinched hysteron I-V curves in resistive switching devices, include the ramp rate in V/s either as a label in the plot or within the caption. This information is crucial to have a grasp into the dynamics of the phenomenon instead of understanding it as a completely stationary process, which is never the case in resistive switching devices.

RESPONSE: Thanks for the suggestion. The sweeping rate is 20 V/s, and this information is added in the caption of Fig.3c in the revised manuscript.

3) For future research, addressing the switching phenomenon by applying pulsed voltage well below 1 ms should be fundamental to understand the true potential of the technology in memory/storage applications.

RESPONSE: We added the corresponding discussion in the revised manuscript to address this issue, pointing out the study with pulsed voltage would be a useful way to further evaluate the intrinsic switching/hydrogenation speed.